# The Impact of COVID-19: The Phenomenological Effect of Burnout on Women in the Nonprofit Sector and Implications for the Post-Pandemic Work World

**Patricia A. Clary * and Patricia Vezina Rose**

Patricia Clary & Associates, Horseshoe Bend, AR 72512, USA
* Correspondence: pat@patriciaclary.com

**Abstract:** Research shows that 67% of the nonprofit sector workforce in the United States are women and worldwide, women account for the majority of employees in the nonprofit sector. Identified as service provider professionals, these women provide the care and nurture of countless people and yet often neglect themselves as they serve others out of passion or a strong work ethic. At the height of the COVID-19 pandemic, service provider professionals responded to an increased demand for programs and services with fewer resources. The increase in the demand for programs and services with a decrease in resources contributed to stress for these workers, leading to the phenomenon of burnout. To address the phenomenon of burnout, we propose that nonprofit organizations need to be systems thinking organizations and consider implications at the organization's micro, mezzo, and macro levels. Three themes emerged from this study, self-care at the micro level, psychological safety at the mezzo level, and reviewed and revised policies and procedures that address the unique needs of women at the macro level. The article considers the nonprofit sector, burnout, and women in the nonprofit sector and its implications for organizations at the micro, mezzo, and macro levels.

**Keywords:** nonprofit; burnout; self-care; well-being; systems-thinking; micro-level; mezzo-level; macro-level; COVID-19

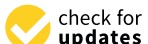



## 1. Introduction

Amidst the ongoing challenges the coronavirus pandemic, COVID-19, brought to the nonprofit sector, nonprofit organizations remain resilient. Staffed by passionate, dedicated professionals, nonprofit organizations seek to reinvent, reorganize, and or reimagine themselves as they emerge from the disruption COVID-19 created globally. A concentrated focus on addressing the phenomenon of burnout for service provider professionals at the three levels of organizational life better positions an organization to remain resilient. However, to do so requires a systems thinking design approach in operationalizing nonprofit organizations. Systems thinking has been gaining momentum in organizational development and change management models. System thinking considers the interconnectedness and interdependence of each component within an organization. Therefore, as the nonprofit community considers how to rebound and rebuild from the rubble of the COVID-19 pandemic, it must consider the micro, mezzo, and macro systems within its organizations. The micro level operates with the individual, the mezzo level occurs at the group or team level, and the macro level happens at the organizational level. Each level is interconnected, whether top-down with policies and procedures that affect the individual and group levels or bottom-up, whereby the individual's intrinsic and extrinsic motivation influences the outcome of group work or the organization's strategic objectives.

In the fast-paced, ever-changing environment of the continued COVID-19 pandemic, the nonprofit sector must consider the price the nonprofit staff and volunteers have paid to fulfill the missional directive of the organization and provide services to meet the needs of those it serves. During COVID-19 and today, the nonprofit sector assisted in helping more

people with fewer resources. Consequently, women in the nonprofit sector were on the frontlines of the pandemic and experienced burnout from stress and a rapidly changing environment. The research for this article led to the phenomenology of burnout experienced by nonprofit service provider professionals. Burnout is defined as a syndrome of emotional exhaustion, depersonalization, and a reduced sense of personal accomplishment in the work of Maslach and Jackson in 1981 [1]. Researchers contributed further to understanding burnout as a response to emotional stress [2], exhaustion, and decreased motivation [3]. The World Health Organization (WHO) defined burnout as "a syndrome resulting from chronic workplace stress that has not been successfully managed" [4]. According to WHO, burnout is an internationally recognized syndrome in which unmanageable workplace stress leads to feelings of exhaustion, cynicism, and negativity about one's job and reduced ability to do that job well [4]. Therefore, this article considers the nonprofit sector, burnout, and women in the nonprofit sector and its implications for organizations at the micro, mezzo, and macro levels.

## 2. Materials and Methods

The researchers wanted to explore the phenomenon of burnout in women in the nonprofit sector and, specifically, how COVID-19 contributed to the phenomenon. The present study sought to answer the following research questions: (1) What does burnout look like in women in the nonprofit sector, and (2) what are the implications for organizational leaders in a post-pandemic work world? Additionally, the researchers have a strong background in theories of organizational leadership, and their understanding of burnout at the organizations' micro, mezzo, and macro levels was essential to provide implications for organizational leaders. The literature review for the article consisted of a systematic search using the keywords burnout, burnout in women in the nonprofit sector, COVID-19 and women in the nonprofit sector, burnout in women, results of burnout in helping professions, the phenomenon of burnout, implications of burnout in women in the nonprofit sector, and burnout in the nonprofit sector. Google Scholar, Emerald Publishing, Steelman Library, Google, and ProQuest were used to identify peer-reviewed articles, reports, blogs, and dissertations addressing keyword searches. The results produced 74 peer-reviewed papers, four dissertations, five nonprofit and business association reports, and 86 articles and blogs from 169 sources. A selective process addressing keyword searches resulted in 42 sources used for the article. The challenge in writing this article was the limited number of sources addressing only women in the nonprofit sector. Therefore, the researchers extrapolated data sets where percentages of the results accounted for 50 percent or more of women. When referenced, the study's results, and the percentage of women are stated in the article. The article presents existing research to make recommendations for the post-pandemic nonprofit world to reduce the burnout that was intensified by all the burden this sector had to endure to meet the needs of the people during the pandemic and based those recommendations upon published data.

### 2.1. The Nonprofit Sector

Through civic engagement, the nonprofit sector furthers social causes to solve complex local and global issues in collaboration with the public, private, and business sectors. A formidable sector, nonprofit organizations play an intricate role in economic and social services delivery systems as they work toward the common good of those they serve [5]. Nonprofit organizations are frequently called upon and play vital roles during times of crisis [6]. During the COVID-19 pandemic, many nonprofit organizations struggled to help their communities while trying to endure the situation themselves. The nonprofit sector is expected to react to social, political, and organizational forces in addition to responding to environmental crises like hurricanes, tsunamis, or other natural disasters. They also react during disruption, for instance, when disruption or displacement occurs due to war or famine. However, COVID-19 was and continues to be unprecedented in the breadth and magnitude of these forces, and the impact on the nonprofit sector remains

unknown. Narrowing to the organizational level, nonprofits across the United States reported operational fallout from the pandemic, including weakened revenue streams and heightened demand for services and support. Early evidence also indicates that not all mission sub-sectors have fared the same, with variations existing according to the subsector's programming, populations served, and even revenue sources [7,8].

A resilient sector, the nonprofit sector is vast, encompassing global concerns like healthcare, the environment, water, humanitarian aid, human rights, human suffering, freedom and democracy, inequity and inclusion, disparity of resources, and sustainability [9]. According to the United Nations, "a civil society organization (CSO) or non-governmental organization (NGO) is any nonprofit, voluntary citizens' group which is organized on a local, national or international level" [10]. GuideStar by Candid registered 14,380 international nonprofit organizations focused on international development and relief services and an additional 8773 focused on international human rights, peace and security, international understanding, and service [11]. There are 185,241 charities in England and Wales as of 10 May 2021. Regarding the gender of volunteers, 66% of the population in England that participated in voluntary activities in 2020 were female [12].

According to a report by the Council on Foundations, non-governmental entities, known collectively as social organizations, reported 810,000 social organizations in The People's Republic of China (PRC) in 2018 [13]. The formal nonprofit organizations, informal nonprofit organizations, and government-organized nonprofit organizations (GONGOs), except for the Red Cross Society of China, which operates as an independent system within The People's Republic of China, are registered at the Ministry/Bureau of Civil Affairs [14]. The nonprofit social organizations and grassroots organizations that have neither official government ties nor the backing of wealthy individuals and or large corporations, deliver the primary care for marginalized groups like people with rare and chronic diseases, pregnant women, the economically disadvantaged, the elderly, and people with disabilities [15]. An extensive search for the number of global nonprofit organizations led to a precursory number of 10 million. However, this number is unsubstantiated, with limited databases to support the cumulative total.

In the United States, nonprofits are non-governmental entities organized to provide services or pursue a mission without earning a profit [16]. Incorporated as tax-exempt entities under section 501(c)3 of the Internal Revenue Code, these organizations represent charitable foundations, private educational institutions, hospitals, healthcare service organizations, social assistance, or service organizations. In addition, child or animal welfare organizations and some types of advocacy organizations are included in the classification. Names synonymous with the nonprofit sector are human service organizations, the third sector, civil society, community-based organizations, nonprofit organizations, non-governmental organizations (NGOs), and voluntary action associations [5].

On 11 July 2022, the United States Internal Revenue Service (IRS) recorded 1,831,723 exempt organizations in the United States. Additionally, 2042 nonprofit organizations were registered in Puerto Rico, with 2197 international nonprofit organizations and 14,003 exempt organizations recorded in the District of Columbia, Washington, D.C., for a total of 1,849,965 exempt organizations registered with the IRS [17]. In the United States, the nonprofit sector received $449.64 billion in charitable contributions in 2019 [18]. Furthermore, the industry contributed $1.4 trillion to the economic base of the United States in the first quarter of 2022 and accounted for more than 12 million jobs in 2016 [19,20]. At the center of the nonprofit sector are women, who comprise 75 percent of the workers in education, healthcare, and social assistance, the industries that encompass most U.S. nonprofits [16].

The nonprofit delivery system is essential in the United States to meet the growing needs of people exacerbated by the global COVID-19 pandemic of 2020. Coming out of the pandemic in 2021, current President Biden reestablished the White House Office of Faith-Based and Neighborhood Partnerships within the President's Executive Office, strengthening the nonprofit sector and its partnership with the federal government [21]. The executive order cited "the global pandemic, a severe economic downturn, systematic racism,

climate crisis, and polarization as reasons to seek civil society partnership to meet such challenges [5]. Abramson (2020) and Feiock and Andrew (2006) contended that nonprofit organizations are valuable partners and conduits in the federal government's delivery of programs and services to meet the growing needs of people in the United States [22,23]. However, new data provide further evidence that the public served by nonprofits continues to be at risk. In the face of the ongoing public health and economic crises, too many nonprofits are still struggling to meet increased service demands, confronting a combination of decreased revenue and expenses that are higher than pre-pandemic contributing to stress and the phenomenon of burnout [24].

Post-COVID-19 research pertinent to the impact of the coronavirus on women in the nonprofit sector associated with social service organizations or direct delivery providers is almost non-existent or limited. While research is conducted in the education, healthcare, and mental health sectors, all of which fall under the umbrella of the nonprofit sector, there remains limited research on the impact on women as service provider professionals. However, in a 2021 report conducted by the Center for Nonprofit Philanthropy and research partners, in a sample size of 2306 direct service providers and community building advocacy, the research showed:

- On average, half of the board members identify as women;
- Sixty-two percent of executive directors are female;
- Forty-nine percent of board chairs are female;
- More than half of the average organization's staff are women;
- Twenty-two percent reported their staff is all women [25].

More research is needed to understand the impact of COVID-19 on service provider professionals, specifically women. For this article, however, what is available to explore and learn from is extractable evidence-based research, results, and implications based on empirical data where more than 50% of the research responses were from women in nonprofit organizations.

The research for this article led to the phenomenology of burnout experienced by nonprofit service provider professionals. The phenomenon of burnout was evident in a systematic literature review that extended across all classifications of nonprofit organizations registered with accrediting agencies. Furthermore, burnout was a phenomenon worldwide as women responded to the COVID-19 global pandemic. Therefore, this paper examines how burnout impacted women in the nonprofit sector during the pandemic and what changes need to take place to address burnout now and in a post-pandemic workplace. The implications will provide nonprofit and business leaders with the knowledge necessary to support women in the nonprofit sector, strengthening civil society. Additionally, the research will help the nonprofit sector understand how COVID-19 reshaped the nonprofit sector's workforce and what is needed to engender the commitment of the nonprofit workforce beyond the pandemic [6].

## 2.2. Burnout and Women in the Nonprofit Sector

The U.S. Bureau of Labor Statistics reported in September 2020 that the COVID-19 recession has been tougher on women, with a disproportionately negative effect on women and their employment opportunities [26]. According to The Independent Sector 30 June 2022, Health of the U.S. Nonprofit Sector, a reported 67.9 percent of the nonprofit workforce in the United States were women [19]. Additionally, this figure jumped to almost 70% in the healthcare industry [19]. In 2020, The Organization for Economic Co-operation and Development (OECD) reported that women were at the center of the global pandemic shouldering much of the burden associated with COVID-19 [27]. In the OECD community, just over 60% of public sector workers are women, and roughly 70% are in Denmark, Finland, Norway, and Sweden (OECD, 2019) [27]. A study of the motivation of volunteers in Bahrain and Bangladesh found that in collectivist Islamic societies, women made up a large percentage of volunteers during the COVID-19 crisis. "The motivations behind the young women volunteering in Bahrain also appear to be associated with a sense of

obligation, a desire to place the interests and benefits of the community and nation before their own, and a willingness to sacrifice self-interest for the greater common good" [28] (p. 14). Additional research showed that omen in the nonprofit sector were also susceptible to a heightened risk of job and income loss, inequities at work, increased violence and abuse, and additional caregiving responsibilities at home [27], all of which contribute to the phenomenon of burnout. As we know today, COVID-19 harmed the health, social, and economic well-being of people worldwide, and at the center of the fight against COVID-19 were women in the nonprofit sector.

During the COVID-19 crisis, most schools worldwide were closed indefinitely. One study showed that 73.5 million children in the United States are under 18. Of these, 30 percent live in single-parent households. The current crisis affects single mothers more significantly. If all schools in the U.S. are closed for an extended period and single mothers cannot work, these children are at risk of living in poverty. There is little room for alternative arrangements in the COVID-19 crisis [29].

Bandali (2020) posited gender stereotypes in NGOs depicted women in the nonprofit sector as self-sacrificing, caring, and nurturing. These perceptions of women impact their emotional and physical health as they take on overwork leading to burnout. In the study of Malaysian women in the nonprofit sector, Bandali found that they worked with little remuneration, were often exhausted, received little accolades, and rarely thought about their care. Furthermore, Bandali found that the working culture mantra, *the work is good-the work always comes first*, leads to women exiting the sector contributed to the phenomenon of burnout [30]. A study of aid workers, where over three-quarters of those that took the survey were female, found that 79% of the 754 respondents stated they had experienced mental health issues. The research results showed little gender differentiation, with half of the contributors reporting they experienced panic attacks, post-traumatic stress disorder, and depression.

Young's (2015) study submitted that staff welfare often took a backseat to taking care of clients [31]. Severe stress and high ideals in helping professionals who sacrifice themselves for others often experience burnout, exhaustion, and inability to cope [31]. A study of 3542 Utah women, of which 65.3 percent of the population worked in classified IRS tax-exempt nonprofit organizations, reported mental decline, burnout, and exhaustion from additional responsibilities in the home [32]. In a study by Kannampallil et al., (2020) on how exposure to frontline healthcare workers contributed to physician trainee stress and burnout, 66% of the resident respondents were female. Kannampallil's study showed that the exposed group experienced perceived stress regarding childcare, reported lower work-family balance, interference with family life, and more difficulty taking time off to attend to personal or family matters. Additionally, stress, burnout, anxiety, depression, and low professional fulfillment from clinical work activities were prevalent, with women trainees more likely to have higher stress levels [33].

In a qualitative study of healthcare workers in Iran, burnout emerged as one of the three main themes [34]. The study showed that increased workload, reduced family relationships, and a lack of motivational factors contributed to burnout. Moreover, in a survey of physicians, law enforcement, and clergy located in two southwestern states in the United States, of which 105 participants were women, results showed emotional exhaustion, depersonalization, and personal accomplishment, and factors of burnout were predictors of low career commitment during the COVID-19 pandemic [35]. The research in this article contributes to a better understanding of how the phenomenon of burnout affects women at the organization's micro, mezzo, and macro levels.

## 3. Results and Recommendations

As researchers interested in organizational leadership, we wanted to understand the phenomenon of burnout in women in nonprofit organizations, report the results, and make recommendations on the research findings. The results and recommendations are presented at an organization's micro, mezzo, and macro levels.

### 3.1. Micro-Level: The Individual

The implication at the micro level is for the organization to consider a woman's emotional, mental, and physical needs in establishing a work–life balance. A work–life balance reduces the stress and burnout experienced by women on the frontline who are vulnerable to a lack of self-care driven by a passion for helping others [30]. These frontline women are known as "helping" professionals [30]. The passion "helping" professionals bring to the workplace is characterized as a self-sacrificing, caring, and nurturing persona that impacts a woman's emotional and physical health. The characteristics of "helping" professionals are often found in service provider professionals, where being passionate reinforces the idea of being selfless and where the wake-up call for self-care is often a serious illness [30]. Self-care can be as simple as providing time for walks during breaks at work or reorganizing office space conducive to relaxation, meditation, or yoga. Women's self-care in the nonprofit sector can begin with designing a personalized self-care regime based on the needs and the work–life balance they seek [30].

A part of self-care is access to free psychological counseling services with regular mental health assessments to benchmark a woman's progress toward a work–life balance that reduces stress and burnout. Self-care can also occur in women-to-women mentoring or nurturing groups where women contribute to the emotional well-being of other women in the organization. Organizations can adopt similar policies like the Accreditation Council for Graduate Medical Education (ACGME). ACGME guidelines provide time off to attend medical appointments, access to mental health services, and flexible work schedules [33]. A micro-level analysis of women's self-care in the nonprofit sector provides an organization with a deeper understanding of how to help the women in their organizations manage every detail of their self-care.

### 3.2. Mezzo-Level: Managers and Leaders

The mezzo level of the organization considers the employees' group life. At this level, research shows that the line manager, mid-level leadership, or group leaders are critical to promoting a healthy work environment for women in the nonprofit sector. To promote a healthy workplace environment at the mezzo level, policies that address the well-care of employees should be reviewed and adopted by managers and leaders. As the stress and growing burnout phenomenon continue post-COVID-19, managers, and leaders who emphasize the importance of an employee's good mental health and create psychologically safe workplaces will help reduce stress and burnout [36]. There is a need for managers and group and team leaders to make time at work for the processing of emotions caused by stress and burnout. Research shows that leaders who role-model their emotional debriefing and are open about their experiences help create an environment for shared experiences that contributes to the well-care of employees [37].

Manager decisions and management styles are critical factors in promoting healthy emotions and psychological safety [36]. A psychological safety net is the beginning of addressing burnout at the mezzo level. Additionally, trust, empathy, and autonomy reflect positively on employees, while management styles of micromanagement and control reflect negatively on employees, especially during a crisis [36]. Moreover, empowerment of workplace decisions, open communications, assurance, and trust in shared goals is crucial to employee psychological safety [38].

### 3.3. Macro-Level: The Organization

At the organization's macro level, the internal and external environmental factors contributing to an employee's well-being and job satisfaction should be well-thought-out. For example, a fallout leading up to the pandemic, coined by Klotz (2021) as The Great Resignation, is a movement cutting across all industries where significant numbers of employees voluntarily resigned from their positions, leaving employers short-staffed [38]. Especially hard hit was the education sector. The National Education Association released a statement in January 2022 that stress and burnout contributed to the great resignation of educators,

with a reported 55% of educators likely to resign or retire earlier than planned. For organizations to remain viable and sustainable, they must address the phenomenon of burnout and environmental influences like The Great Resignation at each level within the organization.

Work-from-home options at the micro level would allow flexibility for women, especially single-parent mothers with increased demands on their time who require work–life balance and psychological well-being to reduce stress and burnout [32]. At the mezzo level, managers and leaders must be empowered to enact processes like rotating work schedules, altering work hours, and revising work expectations [6]. At the macro level, the leadership team of the nonprofit organization can review and change workplace policies and procedures to address the unique needs of women. Critical services for the self-care and well-being of its employees are essential. As is providing opportunities for personal and professional development like lunch and learn programs, online homework resources, support benefits like gym memberships, or programs designed to increase employee attendance and retention [36].

While nonprofit organizations struggle to compete with wages in the for-profit sector, they can offer alternatives to help women protect their pay. At the macro level, organizations can revisit policies to align job descriptions and employee wages, return employee pay to pre-COVID-19 rates, and increase employee pay. Companies can show fairness and consideration during crises by readjusting workload instead of cutting pay [36]. Moreover, companies can focus on organizational policies by investing in training for upskilling and family-friendly workplace initiatives with more childcare support. [32,36]. Organizations offering family-friendly policies have a positive impact on the entire community increasing employee diversity, productivity, and job satisfaction [32]. In addition, organizations can provide awareness to all parents on work–life family programs like The Family Security Act 2.0, which accentuates support for working families and resources available in raising and educating their children [39]. Fundamentally, all policy responses to the crisis must embed a gender lens and account for women's unique needs, responsibilities, and perspectives [27].

An organizational climate survey is another approach to help nonprofit leadership reevaluate their workforce. For example, validated instruments such as the model for job role conflict and ambiguity (Netemeyer et al., 1995) can help organizations understand the needs of the employees in developing policies and procedures that contribute to the self-care and well-being of women in the nonprofit sector [40]. Furthermore, documenting the changes that nonprofit workers have experienced is an essential first step toward understanding COVID-19's impact on the sector's workforce for job role conflict and ambiguity [41]. In addition, organizations should consider recruiting women who have left the workplace during the pandemic and implement longer-term strategies for recruiting women returning to the workplace after career breaks. In another arena, the nonprofit community relies heavily on donor support, and allocating resources to help women is critical. "As women remain highly represented in care professions, it is time that nonprofit/NGO working environments, donors, and larger infrastructures look within organizations and help those who have for so long helped others" [30]. Recognizing women in these organizations and discussing implications for self-care at the micro level, psychological safety at the mezzo level, and revised policies and procedures that address the unique needs of women at the macro level is essential to overcome stress and burnout. The following section offers a discussion and implications for post-pandemic work.

## 4. Discussion and Implications for Post-Pandemic Work

The purpose of this study was to explore answers to two research questions (1) what does burnout look like in women in the nonprofit sector, and (2) what are the implications for organizational leaders in a post-pandemic work world? Research showed burnout is at epidemic proportions in the nonprofit sector. Until the phenomenon is addressed at the micro, mezzo, and macro levels within an organization, it is likely the sector will experience continued burnout coupled with the quiet resignation of many service provider professionals. Moreover, burnout is a global phenomenon across all classifications of

nonprofit organizations registered with accrediting agencies. Furthermore, research showed burnout was more prevalent among women, who comprise a large percentage of the workers in the nonprofit sector. Mothers were more significantly affected as they took on added work with little or no additional remuneration and experienced exhaustion with little accolades and careless abandonment of self-care. Additionally, women reported mental decline, stress, interference with personal lives, family care, anxiety, and depression.

The effects of the COVID-19 epidemic have outlasted the initial waves of the pandemic and have had a significantly negative impact on women in the nonprofit sector. According to UNESCO, at the height of the pandemic worldwide, it was estimated that more than 1.5 billion children were out of school, dramatically increasing the need for childcare. Intensifying the situation was limited care support from grandparents, neighbors, and friends due to the increased risk of contracting COVID-19. In addition, the increased need to care for people with fewer resources contributed to the stress and burnout of employees in the field. In response to COVID-19, nonprofit organizations modified work hours, reduced pay, and changed the delivery of services [41].

According to Maslach & Leiter (2005), two paths to focusing on employee burnout centers on the individual in the organization and the organization itself. We present that addressing factors contributing to the stress and burnout of employees is paramount to the ongoing success and sustainability of the nonprofit sector [42]. Therefore, we argue that a systems thinking design model for managing stress at the organization's micro, mezzo, and macro levels is critical in the post-pandemic environment. As pertinent to this article, systems thinking at the micro level represents the employee. The mezzo level is the leadership, the group or team leaders, and the macro level considers the organization's culture, policies, and procedures. System thinking considers the interconnectedness and interdependence of each component within an organization. Therefore, as the nonprofit community considers how to rebound and rebuild from the disruption of the COVID-19 pandemic, it must consider the micro, mezzo, and macro systems within its organization. The micro level considers the individual, the mezzo level is the group level, and the macro level is the organizational level. Each level is interconnected, whether top-down with policies and procedures that affect the individual and group levels or bottom-up, whereby the individual's intrinsic and extrinsic motivation influences the outcome of group work or the organization's strategic objectives.

A concentrated focus on burnout at the three levels of organizational life better positions an organization to remain resilient as they reinvent, reorganize, or reimagine themselves with a team of dedicated service provider professionals responding to the needs of those they serve. The implications are far-reaching for post-pandemic work as the nonprofit sector considers how to rebuild and rebound from the COVID-19 pandemic.

## 5. Conclusions

The nonprofit sector rose above challenges encountered during the global COVID-19 pandemic. The mission-driven organizations staffed by service provider professionals delivered programs and services essential to the well-being of people around the globe with limited resources and increased demands. However, it was not without cost. Not all nonprofit organizations came through the pandemic unscathed, recognizing that such a response to increased demands and limited resources contributed significantly to the burnout of their employees. In addition, the care and nurturing of people during the COVID-19 pandemic fell primarily to women in the nonprofit sector.

Today, the impact of COVID-19 continues to leave its imprint within the nonprofit sector at the organization's micro, mezzo, and macro levels. The self-care and well-being of women in nonprofit organizations led to the further realization that nonprofit organizations must look at how they provide opportunities to the women within their organizations for self-care. At the mezzo level, organizations must ask themselves how they provide a psychologically safe environment for employees to receive mental health opportunities that address the stress and burnout they are experiencing and how such opportunities contribute

to the well-being of women. Furthermore, at the mezzo level, managers and group and team leaders must have the authority to provide workplace flexibility to women with increased responsibilities. Most critical at the macro level is the review and revision of policies and procedures that are not conducive to reducing stress and burnout in consideration of the unique needs of women in the nonprofit sector. The global pandemic taught us that it is equally vital for the care and nurturing of the women within the organization—as it is—to have a mission-driven organization staffed by service provider professionals.

**Author Contributions:** Conceptualization, P.A.C. and P.V.R.; methodology, P.A.C. and P.V.R.; software, P.A.C. and P.V.R.; validation, P.A.C. and P.V.R.; formal analysis, P.A.C. and P.V.R.; investigation, P.A.C. and P.V.R.; resources, P.A.C. and P.V.R.; data curation, P.A.C. and P.V.R.; writing—original draft preparation, P.A.C. and P.V.R.; writing—review and editing, P.A.C. and P.V.R.; visualization, P.A.C. and P.V.R.; supervision, P.A.C. and P.V.R.; project administration, P.A.C. and P.V.R.; funding acquisition, P.A.C. and P.V.R.; All authors have read and agreed to the published version of the manuscript.

**Funding:** This research received no external funding.

**Institutional Review Board Statement:** Ethical review and approval were waived as this article presents existing research.

**Informed Consent Statement:** Not applicable.

**Data Availability Statement:** Not applicable.

**Conflicts of Interest:** The authors declare no conflict of interest.

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
