# Peer review of "The Impact of COVID-19: The Phenomenological Effect of Burnout on Women in the Nonprofit Sector and Implications for the Post-Pandemic Work World"

_merits, doi:10.3390/merits2040023_

Round 1

Reviewer 1 Report

The paper is interesting, showing the phenomenon of burnout of women in the non-profit sector. However, I have some recommendations to increase the scientific quality of the paper:

In order to enhance the clarity and readability of the paper, the authors must specify in the introduction the originality of the research. I also suggest that the introduction has to contain the objectives, organization and motivations of the research. I also recommend to add methodological part to the article. Even if the article is purely theoretical, it is necessary to define what literary sources the authors rely on and what is the logic behind the whole concept of the article.

The article does not have a separate chapter devoted to the literature review (which I don't consider a problem), but that is why the logic of the individual subchapters needs to be explained. 

As article is organized, it looks more like a state of the art. (I am referring also to part 3, which is supposed to be the “results” part of the article). It is necessary to highly added value of the authors´ research and the article.

Furthermore, between one part to another, I would suggest a small introduction of the next paragraph, to improve the sequencing of the reading. I strongly suggest making a general discussion, highlighting the research results.

Reviewer 2 Report

This is a very interesting review of research on burnout of women in the nonprofit sector and suggestions for addressing this at the micro, meso and macro levels. 

When the author(s) outline the problem in the first half, it is very clear that they are describing the results of research.  This is less obvious in the second half, or the recommendations.  While it seems quite likely that they are reporting on research findings based on the citations, it is not stated in the same way as the first half of the paper.  

Could be some clarification around whether the paper is addressing "helping" professions or non-profit work.  The title suggests non-profit, but helping professions, such as health care, are referenced in the paper.  Of course, health care can be non-profit, it can be for profit, it can be government based.  

Given that the authors are referring to research from a variety of countries, it would be helpful to clarify which country the Family Security Act 2.0 is from (p. 6).

Finally, I was surprised to see the macro level recommendations did not include adjusting compensation or better aligning compensation with job descriptions.  Perhaps the research doesn't support this, but it seems like a reasonable policy/practice to be regularly reviewing compensation and job duties.  

Round 2

Reviewer 1 Report

The authors have incorporated the comments of the reviewers in a relevant way and I consider the article in a modified form suitable for publication.

Author Response

Hi,

Thank you for your kind comment. We appreciate you and your generosity of time and expertise.

Warm regards,

Patricia A. Clary

Patricia Vezina Rose